# Side Effects Reported by Jordanian Healthcare Workers Who Received COVID-19 Vaccines

**DOI:** 10.3390/vaccines9060577

**Published:** 2021-06-01

**Authors:** Osama Abu-Hammad, Hamza Alduraidi, Shaden Abu-Hammad, Ahmed Alnazzawi, Hamzah Babkair, Abdalla Abu-Hammad, Ibrahim Nourwali, Farah Qasem, Najla Dar-Odeh

**Affiliations:** 1College of Dentistry, Taibah University, Al Madinah Al Munawara 43353, Saudi Arabia; oabuhammad@taibahu.edu.sa (O.A.-H.); anazawi@taibahu.edu.sa (A.A.); hbabkair@taibahu.edu.sa (H.B.); Ibrahim_germany@hotmail.com (I.N.); 2School of Dentistry, University of Jordan, Amman 11942, Jordan; 3School of Nursing, University of Jordan, Amman 11942, Jordan; h.alduraidi@ju.edu.jo; 4Comprehensive Amman Healthcare Center, Amman 11192, Jordan; Shadenabuhammad@gmail.com; 5School of Medicine, University of Jordan, Amman 11942, Jordan; abdullah018ju@gmail.com; 6University of Jordan Hospital, Amman 11942, Jordan; farahqasem@outlook.com

**Keywords:** COVID-19, AstraZeneca vaccine, Pfizer-Bionteck vaccine, SinoPharm, side effects, healthcare professionals, physicians, nurses, dentists

## Abstract

Background Distribution of COVID-19 vaccines has been surrounded by suspicions and rumors making it necessary to provide the public with accurate reports from trustworthy experts such as healthcare professionals. Methods We distributed a questionnaire in Jordan among physicians, dentists and nurses who received a COVID-19 vaccine to explore the side effects (SE) they encountered after the first or the second dose of one of three vaccines namely: AstraZeneca Vaxzevria (AZ), Pfizer-BioNTeck (PB), and SinoPharm (SP) vaccines. Results A total of 409 professionals participated. Approximately 18% and 31% of participants reported no SE after the first dose and second dose, respectively. The remainder had mostly local side effects related to injection site (74%). Systemic side effects in the form of fatigue (52%), myalgia (44%), headache (42%), and fever (35%) prevailed mainly after the first dose. These were significantly associated with AZ vaccine, and age ≤ 45 years (*p* = 0.000 and 0.01, respectively). No serious SE were reported. Conclusions We can conclude that SE of COVID-19 vaccines distributed in Jordan are within the common range known so far for these vaccines. Further studies are needed to include larger sample size and longer follow-up period to monitor possible serious and long-term SE of the vaccines.

## 1. Introduction

The long-awaited vaccines of COVID-19 (corona virus disease-2019) were not received by cheers in many areas of the world whether in developed or developing countries where a substantial proportion of populations cast their doubts and suspicions [1]. Many rumors and misinterpretations surrounded the different vaccines to variable degrees.

The devastating pandemic caused by severe acute respiratory syndrome coronavirus 2 (SARS-CoV-2) has been associated with large numbers of cases, mortalities and severe long-term complications influencing individuals and economies.

In Jordan, the first cases of COVID-19 appeared in March 2020, but due to strict measures adopted by the government, the first wave was mild, and outcomes of the pandemic were relatively trivial [2]. However, the country was hit hard by a more aggressive second wave during September 2020. As of mid-April 2021, the 10-million population country had over 672,000 total cases (65,364 cases per million) and 7937 deaths (772 per million) [3]. Jordan was among the countries that had an early start in vaccination campaigns which started in January 2021. Taking into consideration that older adults (aged ≥ 70 years) are at increased risk of severe disease and death if they develop COVID-19 [4], these were prioritized for vaccination. Healthcare workers were also selected for vaccination according to their age, hence, there were several COVID-19 related deaths registered among them.

Although many physicians were reported dead due to infection, the nation was shocked to receive the news of a young female physician who contracted infection and later was deceased due to the infection in late March 2021. Consequently, the Jordanian government intensified the vaccination campaigns and initiated a campaign that specifically targets all healthcare workers regardless their age. A nationwide campaign was established by the Ministry of Health and under a close supervision by armed forces to provide vaccines free of charge to all healthcare personnel.

Three types of vaccines are being administered currently in Jordan according to availability: AstraZeneca Vaxzevria, Pfizer-BioNTeck (PB), and SinoPharm (SP) vaccines. SE to COVID-19 vaccines were, and still are, a controversial issue and a continuously evolving situation. According to Centers of Disease Control and Prevention (CDC), the common adverse effects include pain, swelling and redness at injection site, as well as fatigue, chills, fever, myalgia, headache, and nausea [5].

The vaccination campaign in Jordan was counteracted by another campaign spreading doubts around the efficacy of these vaccines and intensifying fears around their safety profile. Despite the large numbers of COVID-19 cases and the early start of the vaccination campaign in Jordan, there is little data on either clinical outcomes of infection or even the SE of vaccines. It is important at this stage to collect data on these SE of distributed vaccines so as to inform and educate the public on this issue. More importantly, it is essential to collect this data from trustworthy and expert sources who are willing to provide accurate and transparent data that is not affected by geopolitical vaccine wars.

Therefore, we invited Jordanian healthcare professionals including physicians, nurses and dentists to report the SE they encountered after receiving one or two doses of COVID-19 vaccines.

## 2. Materials and Methods

This was a cross-sectional survey utilizing an online questionnaire. The questionnaire was anonymous, and it was composed of eight closed-ended questions on gender, age, occupation, vaccine type, observed side effects and their duration after first dose and second dose. The questionnaire was distributed among three categories of healthcare personnel in Jordan: nurses, dentists and physicians.

The study was ethically approved by the ethical committee, School of Nursing, University of Jordan IRB#PF.21.14.

### Statistical Analysis

IBM-SPSS statistical package for social sciences version 21 (IBM Corp: Armonk, NY, USA) was used to obtain descriptive statistics and significant associations between variables.

Descriptive statistics were reported as frequencies and percentages, means and standard deviations. Cross tabulation with chi square test was carried out to identify significant associations between severity of side effects as the dependent variable and independent variables of vaccine type, gender and age.

## 3. Results

The study sample consisted of 409 healthcare personnel (nurses, dentists or doctors) working in Jordan.

### 3.1. Participants’ Characteristics

Age, gender distribution, profession, and vaccine types received are displayed in Table 1.

### 3.2. Reported Side Effects 

Number of side effects reported after receiving the first dose ranged between 0 and 11 for each participant, with a mean of 3.2 ± 2.7 per participant. Number of side effects reported after receiving second dose ranged between 0 and 9, with a mean of 1.4 ± 2 per participant. Figure 1 presents the reported SE of each vaccine listed according to their frequency and Table 2 and Table 3 present frequency (%) of side effects reported after first and second doses, and statistical significance of their association with various vaccine types after the first dose and second dose separately.

Table 4 presents less frequent side effects that occurred after first or second dose for all vaccines.

### 3.3. Relationship of Severity of Side Effects with Age and Gender and Vaccine Type

Severity was expressed as no reported side effects, local side effects (injection site pain or arm numbness) and systemic side effects (all remaining side effects). Cross tabulation of severity of side effects with type of vaccine following both doses, showed that AZ was found to be significantly associated with more severe side effects, while SP was significantly associated with no side effects and PB was significantly associated with local side effects (*p* = 0.000) (Table 5). Age group but not gender was significantly associated with severity of SE after first dose only (*p* = 0.01), however, no statistically significant association was found between either gender or age group and severity of SE after second dose (*p* > 0.05).

### 3.4. Duration of Systemic Side Effects after Dose 1 in Days

On average systemic side effects lasted for 1.39 ± 1.12 days, with a range of 0–5 days. When the patient reported recovery from severe SE within 24 h of their start, the duration was calculated as 0 days.

## 4. Discussion

We conducted this study to explore the side effects of COVID-19 vaccines among their recipients of healthcare professionals to ensure, as much as possible, the accuracy and credibility of data collected. Physicians, nurses, and dentists were invited to participate in this survey because we think that these professionals are strongly related to the critical health aspects of the pandemic. Physicians and nurses work in the front lines in the combat against the pandemic, whereas dentists are at a high risk to contract the infection due to the intimate and risky nature of dental practice in dealing with highly infectious fluids and aerosols [6].

Until now vaccination campaigns against COVID-19 have been influenced by rumors, suspicions, hesitancy and refusal. There was also exaggeration and over-reporting of adverse effects of vaccines, as some of these effects are normal physiologic processes or developmental anomalies that cannot be related in any way to any drug and definitely not to vaccines. Some reported SE are either normal physiologic processes such as teething, while others could be developmental conditions such as fissured tongue [7]. Monitoring the safety of COVID-19 vaccines is an important and ongoing process that should also be accurate. In the US, Vaccine Adverse Event Reporting System has been implemented as an active surveillance system, during the initial implementation phases of the COVID-19 national vaccination program [8]. A similar system is being adopted in Europe by individual national authorities, in collaboration with the European Centre for Disease Prevention and Control and European Medicine Agency (EMA) [9].

In Jordan at least 524,533 doses of COVID-19 vaccines have been administered as of 15 April 2021, which are enough to vaccinate about 2.6% of the population utilizing the two-dose regimen [10]. Unfortunately, little research has been published on COVID-19 clinical outcomes in Jordan and as far as we know no research has been conducted there on either efficacy or safety of its vaccines so far.

The study included three vaccines and it investigated SE after first and second doses separately because we aimed to collect as many responses as possible, and to explore possible variations in reactions between first and second doses. It is important at this critical stage of the vaccination campaign to reassure vaccine recipients by collecting evidence-based data about the local and systemic SE especially if these effects have a transient or temporary nature which might abolish fears and encourage completion of the two-dose vaccination series [8], and booster doses in the future if need arises. The three vaccines included in this study were PB (an mRNA vaccine), AZ vaccine-Vaxzevria, (adenovirus vector encoding the S glycoprotein of SARS-CoV-2), and SP (inactivated vaccine). PB and AZ vaccines are being distributed on a wide scale and their SE are being continuously documented and reported. On the other hand, there is little information on SP vaccine in terms of its efficacy and safety. Currently, a limited number of countries adopted SP in their vaccination campaigns including Jordan, United Arab Emirates, Bahrain, Egypt and Peru [11]. Hence, it is important to provide data on this particular vaccine as well.

Approximately one in five and one in three did not have side effects after the first and second dose, respectively. The lower rate of side effects in second dose recipients noticed in this study is probably correlated to the variation in numbers of various vaccine recipients among first and second dose recipients. Less than 4% of the second dose recipients (compared to 44% of first dose recipients) have received AZ vaccine which was found to be the one highly associated with side effects. On the other hand, approximately 45% of second dose recipients have received SP vaccine, the one with minimal side effects. Most participants who received SP vaccine did not report any side effects after any of the two doses indicating the weak immunogenic potential of the vaccine. Inactivated vaccines are said to have a good safety profile, however, they need a booster plan to create immune memory [12]. Most participants had local side effects associated with post-injection pain and numbness. Among systemic side effects, fatigue represented the most common symptom after both first and second doses. Other cited side effects included myalgia, headache, fever, arthralgia, and bone pain. According to Centers of Disease Control and Prevention (CDC), the common adverse effects of COVID-19 vaccines include pain, swelling and redness at injection site, as well as fatigue, chills, fever, myalgia, headache, and nausea [5]. A minority of participants reported gastrointestinal side effects (nausea, vomiting, diarrhea), respiratory side effects (dyspnea), and there were some sporadic cases of ear symptoms, facial pain, sleepiness, and diuresis.

PB vaccine SE were reported in literature as fatigue, chills, headache, myalgia, and pain at the injection site. Such adverse events were dose-dependent and were more common after the second immunization [13]. This is consistent with our study wherein PB vaccine was significantly associated with headache, fever, fatigue, myalgia and joint pain after the second dose. This vaccine can also rarely cause anaphylactic shock [14]. According to Medicines and Healthcare Products Regulatory Agency (MHRA), it has been advised to administer the vaccine with caution to individuals who have any history of an allergic reaction to a vaccine, drug or food and especially people who need an adrenaline auto-injector in emergency cases [15]. In comparison to other COVID-19 vaccines it is more temperature sensitive, consequently, it is harder to store and transport [14]. None of our participants reported allergic reaction to PB vaccine.

The most severe side effects were correlated with AZ vaccine (>90% of vaccine recipients), while the PB was significantly associated with severity of local side effects (46% of PB vaccine recipients). It was reported that Vaxzevria had mild adverse reactions including chills, fatigue, headache, fever, nausea, muscle aches, malaise, and painful injection sites within a week post-vaccination [15]. As a prophylactic measure, paracetamol was recommended to reduce these post-vaccination local and systemic reactions [16]. It was noticed that there were not significant differences between males and females in development of systemic side effects, however, participants who were 45 years or younger had significantly more systemic side effects. Vaxzevria was described as being better tolerated in older adults than in younger adults and has similar immunogenicity across all age groups after a boost dose [4]. Most of the reported local and systemic adverse events in the literature were mild to moderate in severity, and it was found that booster vaccination was associated with fewer adverse effects than the first dose of vaccination and reactogenicity reduced with increasing age [4]. Age restrictions in vaccine selection have been applied in several countries. Although the MHRA recommended against age restrictions in AZ vaccine use, the Joint Committee on Vaccination and Immunisation (JCVI) in the UK advised that people younger than 30 should be offered alternative vaccines where available [17]. Currently, several countries inside and outside Europe have put age restrictions to AZ vaccine recipients [17]. If there are going to be age restrictions in Jordan as well, there is a possibility that a large sector of the population will be offered other alternatives because the median age is estimated to be 23.8 years. Of interest are the gastrointestinal symptoms affecting recipients of AZ vaccine. The vaccine was found to be clearly associated with diarrhea, nausea and vomiting. This may be explained by the nature of this vaccine (S glycoprotein) and its influence on the gastrointestinal system, an influence that was previously elucidated for SARS-CoV-2 itself [18]. No serious SE are being reported by our study. According to EMA, 30 cases of thromboembolic events (predominantly venous) had been reported by 10 March 2021, among the approximately 5 million recipients of the Vaxzevria in the European Economic Area [19]. It is recommended that people who have persistent and very severe headaches within the first 2–3 weeks after vaccination get further evaluation. A safety update released by the EMA on 16 April 2021 specified thrombocytopenia as a new common side effect (in <1 in 10 persons) and thrombosis in combination with thrombocytopenia as a new very rare side effect (in <1 in 10,000 persons). Side effects included shortness of breath, chest or persistent abdominal pain, leg swelling, severe or persistent headache, blurred vision, persistent bleeding, and skin bruising or round, pinpoint spots beyond the site of vaccination appearing after a few days [20]. Recently, Østergaard et al. (2021) did a nationwide analysis of population-based data from Denmark to estimate the natural incidence of venous thromboembolism and compare it to that related to Vaxzevria, and there was no difference [21]. In tackling this issue, it is important to address the difference in sociodemographics and lifestyle factors among different nations. In Jordan other possible confounders may exist. The role of several psychological triggers related to the COVID-19 pandemic was suggested in a recent study investigating the development of acute myocardial infarction in Jordanian non-COVID-19 patients [22]. These confounders were found to be lockdown stress, loneliness, unstable income, unemployment, binge smoking, anger and fear of contracting COVID-19 infection [22]. Some participants in this study reported headache, dyspnea, tinnitus and facial pain. Whether this is related to the vaccine needs to be further verified. Two cases of herpes zoster were also reported as a side effect in this study. This infection, as well as several recurrent herpes infections, is a known complication for depressed immunity. The influence of the SARS-CoV-2 on the immune system, and its ability to stimulate recurrent opportunistic viral infections of the skin and mucous membranes are already established [23]. Similarly, it seems that the influence of the vaccine on the immune system should not be underestimated.

Our study included three vaccines, two of them have attracted attention worldwide. SP vaccine has been under the spotlight because of its rather “quiet” profile, while AZ vaccine has been surrounded by noise in several pivotal areas of the world. The results of this study confirmed that SP is really a “quiet” vaccine since it was significantly associated with symptom-free vaccination. However, these findings neither confirm its efficacy, nor exclude its long-term SE. On the other hand, AZ vaccine was found to be most significantly associated vaccine with post vaccination side effects. However, it should be noted that no serious SE were reported, and that recovery in all affected participants was achieved in a duration of days.

It was not the aim of this study to investigate efficacy of vaccines, however, five participants considered COVID-19 infection as a side effect. Contracting COVID-19 infection after being vaccinated is consistent with the less than 100% efficacy rate that the producers of COVID-19 vaccines reported after both doses. Vaccine effectiveness in preventing COVID-19 was reported to be 95% for PB [24] and 70.4% for AZ vaccine [25]. As for SP the picture is not yet clear, with efficacy ranging from 50% to 78% according to trials conducted on this vaccine in some countries [11].

There were limitations in this study owing to the cross-sectional, self-reported design and the possibility that psychological factors may have influenced participants’ perceptions. Another limitation is that the study did not investigate medical status of participants, and how this could have influenced their responses to the vaccines. Therefore, we cannot rule out the influence of medical status as an important confounder in determining vaccine SE. Not all our participants have received the second dose yet because it is to be taken in the near future. It was important to include all those who received one or two doses due to the urgent and ongoing nature of vaccine evaluation. Further, response rate could not be determined because at the time of conducting this study the total number of vaccinated healthcare workers in Jordan was not yet known. Therefore, we included a convenient sample. The main strength of this study is that the study sample was derived from healthcare professionals who are expected to provide transparent information based on their medical and scientific background.

It is recommended that longitudinal surveys are conducted over the coming years to investigate possible long-term SE of vaccines. Alongside these surveys, other surveys should be conducted to examine the efficacy of these vaccines in preventing COVID-19, and the best regimen in booster vaccination.

## Figures and Tables

**Figure 1 vaccines-09-00577-f001:**
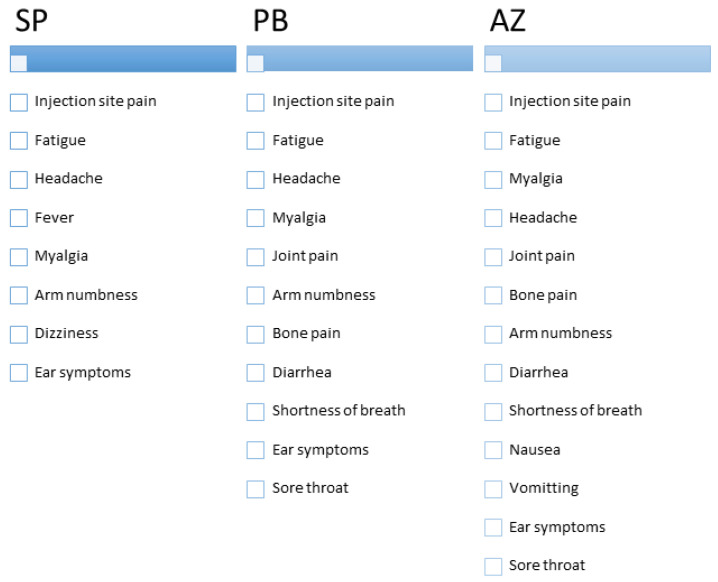
Reported side effects of vaccines arranged according to their frequency. AZ: AstraZeneca Vaxezevira; PB: Pfizer BioNTech vaccine, SP: SinoPharm vaccine.

**Table 1 vaccines-09-00577-t001:** Study sample sociodemographics (N = 409) and types of administered vaccines.

Variable		Dose 1 Recipients Total (409)	Dose 2 Recipients Total (195)
Age (years) (Mean ± SD)		34.99 ± 12.07	39.27 ± 12.79
		Number (%)	Number (%)
Gender	Male	120 (29.3%)	67 (34.4%)
	Female	289 (70.7%)	128 (65.6%)
Profession	Physician	144 (35.2%)	71 (36.4%)
	Dentist	172 (42.1%)	70 (35.9%)
	Nurse	93 (22.7%)	54 (27.7%)
Vaccine type	AZ	179 (43.8%)	7 (3.6%)
	PB	141 (34.5%)	101 (51.8%)
	SP	89 (21.8%)	87 (44.6%)

AZ: AstraZeneca Vaxezevira; PB: Pfizer BioNTech vaccine, SP: SinoPharm vaccine.

**Table 2 vaccines-09-00577-t002:** Side effects (number/percentage) occurring after each vaccine, and statistical significance of their association with various vaccine types after dose 1 (number = 409).

Side Effects	Vaccine Type			Total Number (%)	*p*-Value		
	SP Number (%)	PB Number (%)	AZ Number (%)		SP-PB	SP-AZ	PB-AZ
Side effects No side effects	41 (46.1%) 48 (53.9%)	118 (83.7%) 23 (16.3%)	175 (97.8%) 4 (2.2%)	75 (18.3%)	0.000	0.000	0.000
Fever No Yes	81 (91.0%) 8 (9.0%)	132 (93.6%) 9 (6.4%)	47 (26.3%) 132 (73.7%)	149 (36.4%)	0.462	0.000	0.000
Fatigue No Yes	75 (84.3%) 14 (15.7%)	96 (68.1%) 45 (31.9%)	27 (15.1%) 152 (84.9%)	211 (51.6%)	0.006	0.000	0.000
Myalgia No Yes	83 (93.3%) 6 (6.7%)	111 (78.7%) 30 (21.3%)	36 (20.1%) 143 (79.9%)	179 (43.8%)	0.003	0.000	0.000
Bone pain No Yes	89 (100.0%) 0 (0.0%)	131 (92.9%) 10 (7.1%)	100 (55.9%) 79 (44.1%)	89 (21.8%)	0.010	0.000	0.000
Joint pain No Yes	87 (97.8%) 2 (2.2%)	122 (86.5%) 19 (13.5%)	77 (43.0%) 102 (57.0%)	123 (30.1%)	0.004	0.000	0.000
Headache No Yes	78 (87.6%) 11 (12.4%)	102 (72.3%) 39 (27.7%)	56 (31.3%) 123 (68.7%)	173 (42.3%)	0.006	0.000	0.000
Injection site pain No Yes	57 (64.0%) 32 (36.0%)	34 (24.1%) 107 (75.9%)	16 (8.9%) 163 (91.1%)	302 (73.8%)	0.000	0.000	0.000
Arm numbness No Yes	86 (96.6%) 3 (3.4%)	124 (87.9%) 17 (12.1%)	138 (77.1%) 41 (22.9%)	61 (14.9%)	0.023	0.000	0.012
Diarrhea No Yes	89 (100.0%) 0 (0.0%)	139 (98.6%) 2 (1.4%)	162 (90.5%) 17 (9.5%)	19 (4.6%)	0.259	0.003	0.002
Shortness of breath No Yes	89 (100.0%) 0 (0.0%)	139 (98.6%) 2 (1.4%)	162 (90.5%) 17 (9.5%)	19 (4.6%)	0.259	0.003	0.002
Dizziness No Yes	88 (98.9%) 1 (1.1%)	141 (100.0%) 0 (0.0%)	173 (96.6%) 6 (3.4%)	7 (1.7%)	0.207	0.281	0.028
Vomiting No Yes	89 (100.0%) 0 (0.0%)	141 (100.0%) 0 (0.0%)	173 (96.6%) 6 (3.4%)	6 (1.5%)	No vomiting	0.081	0.028
Nausea No Yes	89 (100.0%) 0 (0.0%)	141 (100.0%) 0 (0.0%)	168 (93.9%) 11 (6.1%)	11 (2.7%)	No nausea	0.017	0.003
Ear symptoms No Yes	88 (98.9%) 1 (1.1%)	140 (99.3%) 1 (0.7%)	177 (98.9%) 2 (1.1%)	4 (1%)	0.742	0.996	0.707
Sore throat No Yes	89 (100.0%) 0 (0.0%)	140 (99.3%) 1 (0.7%)	178 (99.4%) 1 (0.6%)	2 (0.5%)	0.426	0.480	0.865

AZ: AstraZeneca Vaxezevira; PB: Pfizer BioNTech vaccine, SP: SinoPharm vaccine.

**Table 3 vaccines-09-00577-t003:** Side effects (number/percentage) occurring after each vaccine, and statistical significance of their association with various vaccine types after dose 2 (number = 195).

Side Effects Dose 2	VACCINE TYPE			Total Number (%)	*p*-Values		
	SP Number (%)	PB Number (%)	AZ Number (%)		SP-PB	SP-AZ	PB-AZ
Side effects present No side effects	47 (52.8%) 42 (47.2%)	125 (88.7%) 16 (11.3%)	176 (98.3%) 3 (1.7%)	61 (31.3%)	0.000	0.000	0.000
Headache No Yes	78 (87.6%) 11 (12.4%)	104 (73.8%) 37 (26.2%)	177 (98.9%) 2 (1.1%)	50 (25.6%)	0.012	0.000	0.000
Fever No Yes	85 (95.5%) 4 (4.5%)	110 (78.0%) 31 (22.0%)	176 (98.3%) 3 (1.7%)	38 (19.5%)	0.000	0.173	0.000
Fatigue No Yes	74 (83.1%) 15 (16.9%)	82 (58.2%) 59 (41.8%)	178 (99.4%) 1 (0.6%)	75 (38.5%)	0.000	0.000	0.000
Myalgia No Yes	76 (85.4%) 13 (14.6%)	104 (73.8%) 37 (26.2%)	178 (99.4%) 1 (0.6%)	51 (26.2%)	0.037	0.000	0.000
Injection site pain no yes	56 (62.9%) 33 (37.1%)	66 (46.8%) 75 (53.2%)	175 (97.8%) 4 (2.2%)	112 (57.4%)	0.017	0.000	0.000
Numbness injection site No Yes	84 (94.4%) 5 (5.6%)	132 (93.6%) 9 (6.4%)	178 (99.4%) 1 (0.6%)	15 (7.7%)	0.813	0.008	0.003
Joint pain No Yes	86 (96.6%) 3 (3.4%)	118 (83.7%) 23 (16.3%)	178 (99.4%) 1 (0.6%)	27 (13.8%)	0.003	0.074	0.000
Diarrhea No Yes	88 (98.9%) 1 (1.1%)	137 (97.2%) 4 (2.8%)	178 (99.4%) 1 (0.6%)	6 (3.1%)	0.386	0.613	0.103
Shortness of breath No Yes	87 (97.8%) 2 (2.2%)	133 (94.3%) 8 (5.7%)	179 (100.0%) 0 (0.0%)	10 (5.1%)	0.215	0.044	0.001
Bone pain No Yes	85 (95.5%) 4 (4.5%)	122 (86.5%) 19 (13.5%)	177 (98.9%) 2 (1.1%)	25 (12.8%)	0.027	0.078	0.000

Three participants had COVID-19 infection after the first dose of PB vaccine and two had the infection after the AZ vaccine, however, the *p* value was insignificant (*p* = 0.354). Additionally, two had the infection after the second dose of the SP vaccine and one after the PB vaccine, however, the *p* value was not significant (*p* = 0.127). AZ: AstraZeneca Vaxezevira; PB: Pfizer BioNTech vaccine, SP: SinoPharm vaccine.

**Table 4 vaccines-09-00577-t004:** Frequency (%) of less frequent side effects after dose 1 and dose 2 of various vaccines.

Side Effect		Vaccine Type			Total N = 409
		SP (N = 89) N (%)	PB (N = 141) N (%)	AZ (N = 179) N (%)	
First dose	Chest pain	0	0	1 (0.6%)	1 (0.2%)
	Common cold	1 (1.1%)	0	0	1 (0.2%)
	Cough	0	0	1 (0.6%)	1 (0.2%)
	Herpes zoster	0	1 (0.7%)	0	1 (0.2%)
	Loss of smell	0	1 (0.7%)	0	1 (0.2%)
	Lower back pain	0	0	1 (0.6%)	1 (0.2%)
	Palpitations	0	0	1 (0.6%)	1 (0.2%)
	Redness and swelling (injection site)	0	1 (0.7%)	0	1 (0.2%)
	Sleepiness	1 (1.1%)	0	0	1 (0.2%)
	Thirst	0	0	1 (0.6%)	1 (0.2%)
	Urticaria	0	0	1 (0.6%)	1 (0.2%)
	Burning pain in head	0	1 (0.7%)	0	1 (0.2%)
	Loin pain	0	0	1	1 (0.2%)
Second dose	Cough	1 (1.1%)	0	0	1 (0.2%)
	Diuresis	0	2 (1.4%)	0	2 (0.5)
	Herpes zoster	1 (1.1%)	0	0	1 (0.2%)
	Sleepiness	1 (1.1%)	0	0	1 (0.2%)

**Table 5 vaccines-09-00577-t005:** Cross tabulation of severity of side effects after doses 1 and 2 with age groups, gender and vaccine type.

Severity of Side Effects after Dose 1	No Side Effects	Local	Systemic	*p*-Value
Age (years) ≤45 >45	45 (15.8%) 2 (30.3%)	70 (24.6%) 19 (21.3%)	170 (59.6%) 43 (48.3%)	0.010
Gender Male Female	23 (22.1%) 49 (18.1%)	3 (28.8%) 59 (21.9%)	51 (49.0%) 162 (60.0%)	0.154
Vaccine type SP PB AZ	48 (60.0%) 21 (16.4%) 3 (1.8%)	20 (25.0%) 59 (46.1%) 10 (6.0%)	12 (15.0%) 48 (37.5%) 153 (92.2%)	0.000
Severity of side effects after dose 2				
Age (years) ≤45 >45	37 (28.9%) 24 (35.8%)	20 (15.6%) 14 (20.9%)	71 (55.5%) 29 (43.3%)	0.266
Gender Male Female	22 (32.8%) 39 (30.5%)	8 (11.9%) 26 (20.3%)	37 (55.2%) 63 (49.2%)	0.340
Vaccine type SP PB AZ	42 (48.3%) 16 (15.8%) 3 (42.9%)	19 (21.8%) 15 (14.9%) 0 (0.0%)	26 (29.9%) 70 (69.3%) 4 (57.1%)	0.000

AZ: AstraZeneca Vaxezevira; PB: Pfizer BioNTech vaccine, SP: SinoPharm vaccine.

## Data Availability

Data can be obtained from corresponding author upon request.

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
