# Peer review of "Side Effects Reported by Jordanian Healthcare Workers Who Received COVID-19 Vaccines"

_vaccines, 2021, doi:10.3390/vaccines9060577_

Round 1

Reviewer 1 Report

The article explores the side effects after the first or the second dose of one of three vaccines (AZ; PB; SP) by administering a questionnaire to 409 professionals.

Despite the efforts, the article does not present results not already known in the literature and not already reported by vaccine cards.

The introduction is interesting but only traces the pandemic trend in Jordan. It is not conceptually related to the objectives of the study and the experimental part of the article.

The objectives of the study are not well described in the text probably because it is a survey of something that is already known through a questionnaire. It is practically a confirmation on a small sample of something that is already known on more numerous samples.

Even the discussion, although interesting, contains numerous general data without going into depth on the results.

The bibliography is very limited and does not present the other similar experiences already present in the literature. It, as well as the text, also has some formatting errors.

Author Response

Reply file was uploaded

Reviewer 2 Report

The authors did an intense study on the selected topic. The text is generally well written and comprehensive to the readers. The reviewer has very few minor suggestions. 

  1. Providing a simple representative diagram of very important major and minor side effects from selected vaccines is more easy and effective for the readers.
  2. Each section has too many short paragraphs, I would recommend to merge some of those paragraphs.

The authors used sufficient references.

Author Response

Reply file uploaded

Reviewer 3 Report

I believe that this is a valuable contribution to the literature on the side effects associated with the various vaccines, especially as it provides one of the few head-to-head comparisons of three of the most frequently used vaccines. My reservations mainly concern methodology, should be easy to address, and are unlikely to alter the overall conclusions of the paper.

1) Were the participants screened for pre-existing disease at the time of vaccination (e.g., being treated for an infection, autoimmune disease, heart disease, diabetes)? If not, this should be listed as a limitation of the study, since each of these is a known risk associated with severity of COVID-19 and might be expected to influence vaccination as well. Moreover, this medical cohort may be unusually healthy compared with the general population and especially those at highest risk for severe COVID-19.

2) The authors mention that data collected included standard deviations, etc. but I do not see any SD in any of the Tables.  Either the Methods section should delete mention of the SD or they should be included in the Tables where appropriate.

3) A single P value for the comparisons of the three vaccines does not seem to be appropriate or revealing. The way the P values are currently calculated does not reveal which vaccine(s) is/are responsible for any significant differences. The data for each vaccine (in each category) should be compared pairwise with each of the other vaccines and a P value calculated for each pairwise comparison. Such a pairwise comparison will permit the reader to quickly identify which vaccine(s) is/are significantly different from the other(s).

4) Were data available on how many vaccinees refused the second dose of their vaccine? If so, please provide. If not, this should be listed as a limitation of the study.

5) Is it possible to know what was the rate of response to the survey? In other words, how many health care professionals had access to the survey and what percentage of those completed the survey? This is important because it is possible that the results are skewed by vaccinees who experienced greater-than-average side effects wanting to provide negative feedback; or, alternatively, it might be skewed by those who had less-than-average numbers of side effects wanting to provide positive feedback. Again, this difficulty may be listed as a limitation of the study.

RESULTS:

1) The data in Table 2 concerning rate of side effects reported is not consistent with the rates reported in Tables 3 and 4.  Table 2 reports that fewer vaccinees experienced side effects after the second dose than the first, but the rates of side effects reported in Tables 3 and 4 increased for all vaccines from the first dose to the second. Notably, all of the studies of which I am aware have also reported increased rates of side effects (as well as severity) after the second dose so if Table 2 is correct, the lowered rate of side effects after the second dose needs to be addressed vis-à-vis this broader literature. If, however, Tables 3 and 4 are correct, then Table 2 needs to be corrected and the mention of the side-effect rate in the text corrected in several places.

Author Response

Reply file uploaded
